# Urine metabolomic changes in cats with renal disease or calcium oxalate uroliths

Dennis E. Jewell[ID][1]*, Selena K. Tavener[2], Regina L. Hollar[2], Kiran S. Panickar[2]

**1** Department of Grain Science and Industry, Kansas State University, Manhattan, United States of America, **2** Science & Technology Center, Hill's Pet Nutrition, Inc., Topeka, Kansas, United States of America

* djewell@ksu.edu

## Abstract

Two common problems in the urinary system of cats are renal disease (RD) and calcium oxalate (CaOx) stones. The objective of this study was to assess urine metabolomic parameters of cats with these diseases to determine metabolic abnormalities and differences between the groups. The urine metabolic profile for each cat was determined, along with their lifetime history of stone incidence and renal disease. In order to reduce the data for analysis, factor analysis/factor loading was used allowing statistical hypothesis testing and the selection of significant metabolites from 488 identified metabolites. A total of 42 cats were used (19 healthy, 12 with CaOx stones and 11 with renal disease). Urine from the cats were tested multiple times (mean = 4.6) therefore cat ID was used as a random variable. All analytes were expressed as a ratio to creatinine in order to compensate for differences in water intake. Principal components analysis was used as the method of factor extraction resulting in six factors that differed between groups. Factors 1, 2, and 5 were elevated in healthy cats and depressed in RD and CaOx cats. These factors had several analytes that are known to be elevated in the serum of cats with CaOx stones (i.e., 7-methylguanine, erythritol, pseudouridine, N1-methylinosine). Factor 5 was elevated in healthy cats containing six phenyl moieties as well as p-cresol sulfate. There were two factors which were increased in CaOx cats. Factor 12 was increased when compared to healthy cats and contained three purine nucleic acids (inosine, xanthine and hypoxanthine) as well as 3-hydroxybutyrate while factor 23 was elevated and the only factor that contained phospholipids. These results show that urine is not simply reflecting circulating concentration changes observed in cats with CaOx stones and RD but rather also gives insight into the functional kidney changes associated with these diseases.

**Data availability statement:** All relevant data are within the manuscript and its Supporting Information files which now includes the raw data.

**Funding:** This research was funded by Hill's Pet Nutrition, Inc., Topeka, KS, USA (http://www.hillspet.com/our-company.html). Accessed April 27, 2024 which also supplied the excellent technical support of the animal care technicians. The funders had no role in study design, data collection and analysis, decision to publish. The words and conclusions are uniquely the authors and not the funding agency.

**Competing interests:** Three of the authors have an affiliation (K.S.P., S.K.T., and R.L.H) and another formerly had an affiliation (D.E.J.) to the commercial funders of this research, as employees of Hill's Pet Nutrition, Inc which sells food to aid in the management of renal disease and urolithiasis. The funder provided support (in the form of salaries, and access to the colony bioarchived samples), but did not have any additional role. The funders had no role in the design of the study; in the collection, analyses, or interpretation of data; in the writing of the manuscript; or decision to publish. This does not alter our adherence to PLOS ONE policies on sharing data and materials.

## Introduction

It has been reported that cats with non-obstructive kidney stones have a reduced lifespan when compared to geriatric stone-free cats [1]. The actual incidence of kidney stones is unknown; however, urolithiasis accompanies 15 to 20% of cats with clinical signs of lower urinary tract disease [2,3].

There is a variation in the reported composition of uroliths from predominantly calcium oxalate (CaOx; 68.8%) followed by struvite (24.2%) in the Netherlands [4] to a study in Brazil which reported predominantly struvite containing stones (54.4%) followed by CaOx (38.1%) [5]. In a comprehensive study evaluating >11,000 feline stones from Canadian cats from 1998 to 2008 there were 49.1% CaOx stones and 43.0% struvite stones [6]. Nutritional management can result in dissolution of some stones as well as also reducing the risk of stone recurrence [7]. It has been shown that struvite dissolution is highly effective with struvite urolithiasis being dissolved within 30 to 70 days, and this mitigation food is suitable as a maintenance food for long-term prevention of feline struvite urolithiasis [8]. In the study reported here the preponderance of uroliths observed in the colony are CaOx and this stone was required for the entrance of the cat into the CaOx stone forming category.

The treatments that aid in the management of renal disease and calcium oxalate stone formation are an ongoing area of research. Cats with renal disease were shown to have less uremic episodes and renal related deaths when fed a food that had controlled concentrations of protein and phosphorus as compared to a normal adult maintenance food [9]. Cats with an early indication of renal insufficiency (elevated SDMA) were more likely to have stable renal function when compared to the controls. The food fed to these cats with early renal insufficiency contained highly available (but not elevated in concentration) protein, elevated fish oil, antioxidants (C and E), L-carnitine and added vegetables [10]. Active research in cats with RD has included modulated mitochondrial biogenesis. Enhancing mitochondrial biogenesis with sildenafil citrate had a beneficial effect on glomerular filtration [11]. Regarding CaOx stone formation, in most cats with CaOx uroliths, the factors causing CaOx stone formation are unknown [12]. There is a genetic predisposition for CaOx uroliths with increased incidence reported in some breeds [11,13]. The Domestic Shorthair cats had a reduced incidence in compared to many breeds (4). It has been reported that there are a number of risk factors for CaOx stone formation as previously stated [14] "feeding urine-acidifying diets, feeding a single brand of cat food without providing additional foods or table scraps, maintaining cats in an indoor-only environment, and being of the Persian breed". It has been shown that increasing dietary protein resulted in an increase in urinary Ca concentrations, renal Ca and Ox excretion and urinary RSS CaOx which may be result in an increased development of CaOx uroliths in cats [15]. Increasing water consumption resulted in an improvement of risk factors associated with CaOx stone formation [16], Cats with reduced renal function had increased concentrations of circulating creatinine, blood urea nitrogen and phosphorus as well as increased calcium fractional excretion with high salt food [17]. Although feeding high salt to healthy cats does not impair renal function [18]. It has

been shown that increasing the dietary concentration of long-chain polyunsaturated fatty acids lowers the risk of urine calcium oxalate stone formation [19]. It is clear that there is a genetic component to CaOx stone formation includes changes in the AGXT2 gene [20]. However, it is not known what specific metabolomic changes are associated with CaOx stone formation as compared to cats with RD or healthy cats. It is possible that by evaluating these small molecule changes in urine one can identify potential biomarkers of the disease and elevate specific dietary manipulations that would aid in the management of these diseases. This evaluation is the subject of this study with the results showing that urine is different between healthy cats and between these diseases as well as showing possible changes that could be made to identify and aid in the management of these diseases.

## Materials and methods

All study protocols were reviewed and approved by the Institutional Animal Care and Use Committee, Hill's Pet Nutrition, Inc., Topeka, KS, USA. All methods were performed following the relevant guidelines and regulations of the National Research Council of the USA The Institutional Review Board of Hill's Pet Nutrition reviewed and approved the use of the pets described in this experiment. The three approvals were as follows: CP117.0.0.0-A-FU-ADH-MULTI-1-MULTI (January 1, 2010), CP555.0.0.0-A-F-UADH-MULTI-1-MULTI (January 1, 2013), CP678.0.0.0-A-F-U-ADHHILLS-1-MULTI (January 1, 2016).and complied with the National Institutes of Health Guide for the Care and Use of Laboratory Animals [21]. A colony of random bred domestic shorthair cats with indoor runs and access to porches with sunlight were housed in groups. Cats also had exposure to natural light that varied with seasonal changes.. Bioarchive data were used with all cats always living at the Hill's Pet Nutrition, Inc. feline colony.

### Assignment to treatments

Treatment classification was done after the end of life. The CaOx stone forming group contained those cats with CaOx stones in the kidney (the stones analyzed as CaOx). The renal group contained cats with renal disease established by circulating SDMA and creatinine concentrations, urine specific gravity and the evaluation of the kidney. The criteria for the renal disease group was an SDMA over 18 μg/dl, creatinine concentration over 1.6 mg/dl, urine specific gravity under 1.030. Cats could also be assigned based on signs of structural renal disease. Cats with analyzed CaOx stones were not included in the RD group. There were no signs of RD or CaOx stones in the cats assigned to the healthy kidney group.

### Urine metabolomics

Urinary samples were collected throughout life in annual physical exams and at the end of life. Urine was collected from each cat and stored at -80 °C until analyzed. Analysis of urine metabolomic profiles was performed by a commercial laboratory (Metabolon, Morrisville, NC, USA) as previously described [22]. Briefly, the urine sample was split and run on gas chromatography and liquid chromatography mass spectrometer platforms in randomized order. Gas chromatography (for hydrophobic molecules) and liquid chromatography (for hydrophilic molecules) were used to identify and provide relative quantification of small metabolites present in the urine samples. Endogenous biochemicals included amino acids, peptides, carbohydrates, lipids, nucleotides, cofactors, vitamins and other biological compounds. The compound identification was through extracting raw data, peak-identified, and quality control using proprietary hardware and software. Compounds are identified by comparison to library entries of purified standards or recurrent unknown entities. The identification is based on a library of authenticated standards that contains the retention time/index, mass to charge ratio (RI), and chromatographic data (including MS/MS spectral data) on all molecules present in the library. Furthermore, biochemical identifications are based on three criteria: retention index within a narrow RI window of the proposed identification, accurate mass match to the library +/- 10 ppm, and the MS/MS forward and reverse scores. MS/MS scores are based on a comparison of the ions present in the experimental spectrum to ions present in the library entry spectrum.

## Statistical methods

Statistical analyses were performed in SAS version 9.4 (SAS Institute, Cary, NC, USA). This was an exploratory study using 488 urine metabolites in the analysis. In many of these metabolites there was a correlation in the concentrations. This is expected as the samples contain many compounds that are precursors of another compound, are intermediate products in the same system, or similar end products for other biological processes. For appropriate use of this correlation, factor analysis was used as a data reduction method [23,24]. For this study, principal components analysis was used as the method of factor extraction. The resulting patterns were then rotated using a varimax rotation to aid interpretation. The advantage of a varimax rotation is all the factors remain orthogonal (independent) and each metabolite tends to load highly on only one factor, making it easier to interpret the factor patterns. Factor analysis was performed on the correlation matrix so that unequal variances among the metabolites did not unduly influence the resulting factor patterns. Using the correlation matrix, each metabolite has a variance of 1. To reduce the number of factors that needed to be interpreted, only factors with eigenvalues greater than 6 were retained for rotation. An eigenvalue of 8 indicates that a factor accounts for at least 1% of the total variation in the data. Using this approach, the serum metabolites were reduced to 23 factors. The resulting factors were then analyzed using ANOVA as described above as classification (RD, CaOx or healthy) as the fixed effect in the model. Factor means are linear combinations of all metabolites, with higher coefficients or weights for metabolites that load strongly on that factor, and small coefficients or weights for metabolites that are only weakly or not associated with that factor. Because the serum metabolic data is median normalized, a negative mean indicates that the mean level of the metabolic factor is below the median while a positive factor reflects a mean level above the median. Because this is an exploratory study, the unadjusted $p \leq 0.05$ for each factor was used as a cutoff criterion.

## Results

### Pet information

There were 42 cats used in this study with no effect of classification on pet age as shown in Table 1.

The 11 RD cats consisted of 8 female/spayed and 3 male/neutered cats. The 12 CaOx cats were 3 female spayed and 9 neutered while the 19 healthy cats were 10 female/spayed and 9 neutered. This study was a historical evaluation of urine that had been collected by cystocentesis. The average number of metabolomic samples per cat was 4.6.

### Urine metabolomics

The metabolites are listed (Means and standard errors) as a ratio to creatinine concentration in supplementary S1 Table. The eigenvalues and percent of the total variance explained by each factor is shown in supplementary S2 Table. Of the 23 factors that qualified through the statistical cut-offs six achieved the $p \leq 0.05$ level of statistical significance of differences between treatments. The statistical analysis of these factors is shown in Table 2.

The analyte makeup of each of the above factors is shown in Table 3, the complete factor loadings are shown in S3 Table. Factor loadings range between 1 and −1. These factor loadings may be considered similar to correlations between the specific analyte and the central tendency of the factor. A positive value approaching 1 indicates a strong positive

**Table 1. Subject number and average age of cats when sampled.**

| Health | | Age | | |
|---|---|---|---|---|
| Status | n | Mean | SE | Prob > F |
| RD | 11 | 11.56 | 1.31 | 0.2874 |
| CaOx | 12 | 9.16 | 0.84 | |
| Healthy | 19 | 9.41 | 0.72 | |

**Table 2. Factors which were influenced by classification.**

| Variable | Condition | n | Mean | SE | Comparison | Difference | SE | Pr>|t| |
|---|---|---|---|---|---|---|---|---|
| Factor 1 | RD | 44 | −0.69 | 0.18 | RD vs Healthy | 1.07 | 0.23 | <.0001 |
| | CaOx | 54 | −0.19 | 0.17 | CaOx vs Healthy | 0.57 | 0.21 | 0.0112 |
| | Healthy | 97 | 0.38 | 0.13 | RD vs CaOx | −0.50 | 0.25 | 0.0529 |
| Factor 2 | RD | 44 | −0.28 | 0.20 | RD vs Healthy | 0.50 | 0.24 | 0.0428 |
| | CaOx | 54 | −0.16 | 0.18 | CaOx vs Healthy | 0.38 | 0.23 | 0.1009 |
| | Healthy | 97 | 0.23 | 0.14 | RD vs CaOx | −0.12 | 0.27 | 0.6567 |
| Factor 5 | RD | 44 | −0.21 | 0.21 | RD vs Healthy | 0.44 | 0.26 | 0.0923 |
| | CaOx | 54 | −0.31 | 0.19 | CaOx vs Healthy | 0.55 | 0.24 | 0.0301 |
| | Healthy | 97 | 0.24 | 0.15 | RD vs CaOx | 0.11 | 0.28 | 0.7018 |
| Factor 12 | RD | 44 | 0.09 | 0.21 | RD vs Healthy | −0.26 | 0.26 | 0.3189 |
| | CaOx | 54 | 0.32 | 0.19 | CaOx vs Healthy | −0.50 | 0.25 | 0.0507 |
| | Healthy | 97 | −0.17 | 0.15 | RD vs CaOx | −0.24 | 0.29 | 0.4109 |
| Factor 14 | RD | 44 | −0.43 | 0.21 | RD vs Healthy | 0.58 | 0.26 | 0.0338 |
| | CaOx | 54 | 0.15 | 0.20 | CaOx vs Healthy | 0.00 | 0.25 | 0.9853 |
| | Healthy | 97 | 0.14 | 0.15 | RD vs CaOx | −0.58 | 0.29 | 0.0515 |
| Factor 23 | RD | 44 | −0.24 | 0.25 | RD vs Healthy | 0.18 | 0.32 | 0.5776 |
| | CaOx | 54 | 0.46 | 0.24 | CaOx vs Healthy | −0.52 | 0.30 | 0.0936 |
| | Healthy | 97 | −0.06 | 0.19 | RD vs CaOx | −0.70 | 0.35 | 0.0512 |

relationship while negative loadings show a negative correlation which as it approaches 1 shows an increasingly strong negative relationship.

Factor 1 was reduced both by the presence of stones and by RD with the RD cats having concentrations reduced as compared to either healthy or cats with CaOx stones. Therefore, the analytes that were positively loaded in this factor were decreased in the urine of cats with RD or CaOx stones. Factor 1 was weighted in nucleic acid components (9 analytes) that were all positively loaded showing a decrease in urine output of these compounds in cats with CaOx and RD. There was a decrease in factor 2 in cats with RD as compared to healthy cats. This is the result of a concentration decrease in the nine positive loading analytes, of which many are ferulic related (ferulic acid 4-sulfate, 2,6-dihydroxybenzoic acid, dihydroferulic acid sulfate and dihydroferulate). This factor also shows a negative loading and thus a concentration increase in the purine nucleic acid metabolite allantoic acid. Factor 5 was decreased in CaOx stone formers and to a lesser degree in cats with RD. This factor was significantly weighted with phenyl containing moieties (six) with two indole containing compounds as well as p-cresol sulfate. Factor 14 was reduced in RD cats with very similar concentrations in CaOx and healthy cats. This factor contained a pyrimidine nucleic acid (2'-deoxyuridine) and a number of amino acid related compounds (2-aminoadipate, methylmalonate, propionylglycine and tryptophan). Two factors (12 and 23) were elevated in cats with CaOx stones. In factor 12 the elevation was in comparison to healthy cats. This factor contained three purine nucleic acids (inosine, xanthine and hypoxanthine) as well as 3-hydroxybutyrate. Factor 23 was elevated in CaOx cats as compared to cats with RD with the factors 1-palmitoyl-2-linoleoyl-GPC (16:0/18:2), palmitoyl sphingomyelin (d18:1/16:0), valylleucine, S-carboxymethyl-L-cysteine and guanidinosuccinate all loading positively showing a concentration increase in CaOx stone forming cats.

## Discussion

These analytes are expressed as ratios to creatinine in order to reduce the variation resulting from urine dilution. As such, increasing values of a given analyte as compared between groups is associated with a change in urinary output of that analyte as compared to creatinine output. This allows a comparison without the difference associated with water intake

**Table 3. Feline urine metabolite factor loadings of statistically significant factors.**

| Factor 1 | |
|---|---|
| **Metabolite Name** | **Loading** |
| 7-methylguanine | 0.898 |
| 3-(3-amino-3-carboxypropyl)uridine | 0.884 |
| 1-methylguanosine | 0.867 |
| 4-octenedioate | 0.857 |
| biopterin | 0.854 |
| erythritol | 0.854 |
| adenosine 3',5'-cyclic monophosphate (cAMP) | 0.845 |
| 1-methyladenosine | 0.835 |
| O-sulfo-L-tyrosine | 0.835 |
| isoputreanine | 0.833 |
| N3-methyluridine | 0.819 |
| pseudouridine | 0.815 |
| 5,6-dihydrouridine | 0.810 |
| N-acetylglucosamine 6-sulfate | 0.809 |
| N1-methylinosine | 0.809 |
| **Factor 2** | |
| **Metabolite Name** | **Loading** |
| ferulic acid 4-sulfate | 0.831 |
| 2,6-dihydroxybenzoic acid | 0.824 |
| indoleacetylaspartate | 0.781 |
| N6-carboxymethyllysine | 0.748 |
| 4-hydroxycinnamate sulfate | 0.745 |
| dihydroferulic acid sulfate | 0.727 |
| indole-3-carboxylate | 0.727 |
| 4-acetamidobenzoate | 0.722 |
| dihydroferulate | 0.715 |
| 1-methyl-5-imidazoleacetate | −0.319 |
| isocitrate | −0.321 |
| N-glycolylneuraminate | −0.327 |
| allantoic acid | −0.341 |
| **Factor 5** | |
| **Metabolite Name** | **Loading** |
| phenylacetyltaurine | 0.755 |
| indoleacetylglycine | 0.725 |
| phenylacetylglycine | 0.709 |
| phenylacetylglutamine | 0.687 |
| p-cresol sulfate | 0.678 |
| phenylacetylglutamate | 0.673 |
| N-acetylhistamine | 0.653 |
| indoleacetylglutamine | 0.651 |
| 2-piperidinone | 0.636 |
| valerylphenylalanine | 0.634 |
| 4-acetylphenyl sulfate | 0.618 |

*(Continued)*

**Table 3.** (Continued)

| Factor 12 | |
|---|---|
| **Metabolite Name** | **Loading** |
| dibutyl sulfosuccinate | 0.782 |
| hypoxanthine | 0.781 |
| beta-guanidinopropanoate | 0.718 |
| hexanoylglycine (C6) | 0.708 |
| inosine | 0.708 |
| 3-hydroxybutyrate (BHBA) | 0.687 |
| valylleucine | 0.686 |
| isocaproylglycine | 0.685 |
| xanthine | 0.505 |
| **Factor 14** | |
| **Metabolite Name** | **Loading** |
| 2'-deoxyuridine | 0.529 |
| 2-aminoadipate | 0.508 |
| 2R,3R-dihydroxybutyrate | 0.499 |
| methylmalonate (MMA) | 0.470 |
| propionylglycine (C3) | 0.455 |
| tryptophan | 0.401 |
| galactonate | −0.313 |
| N-acetylmethionine sulfoxide | −0.315 |
| S-adenosylmethionine (SAM) | −0.321 |
| 5-methylthioribose | −0.324 |
| isocitrate | −0.390 |
| gulono-1,4-lactone | −0.406 |
| **Factor 23** | |
| **Metabolite Name** | **Loading** |
| 1-palmitoyl-2-linoleoyl-GPC (16:0/18:2) | 0.541 |
| palmitoyl sphingomyelin (d18:1/16:0) | 0.474 |
| valylleucine | 0.450 |
| S-carboxymethyl-L-cysteine | 0.433 |
| guanidinosuccinate | 0.432 |
| indoleacetate | −0.311 |

and the subsequent urinary dilution of that compound. This gives the analysis evaluation a comparative measure in that if the creatinine output is reasonably the same between groups, then the ratio of analyte to creatinine shows a differential excretion of that compound. This is somewhat similar to a factional excretion of the urinary analyte in question which has been previously described (1). This handling of the urinary analyte data as well as the samples being collected over the lifetime of the pet and the different statistical approach in this study may explain the significant differences found in this study as compared to the lack of metabolomic differences between cats with RD and healthy cats previously reported [25].

This study contains a large number of response variables and as such there is a significant need to analyze the data so as to not conclude changes that are the result of normal random variation are biologically relevant. To maximize the understanding of these metabolites this analysis used the statistical relationships between the individual analytes to form

factors. Therefore, the individual factors are an unbiased way to form groups of analytes that are all responding in a similar fashion. This avoids overfitting of the data while it is also unbiased as to the biology of the analytes loading. However, this allows a statistical grouping which has grouped many biological compounds that are related to each other which are also changing concentrations in a similar fashion. For example, the high number of nucleic acid compounds that loaded in Factors 1 and 12 reflect similar biological responses which the statistical approach grouped and highlighted. These two factors are statistically significant while being different in their respective analyte changes with one being elevated and the other declining (both showing differences in the CaOx group as compared to healthy pets). These changes in the compounds making up the factors that were different is in comparison to some factors and compounds which are known to be uremic toxins and were not changed. For example, urea and the dimethyl arginine compounds which are renal toxins known to be elevated in the blood with renal decline [26,27], which when expressed here in the ratio to creatinine in the urine were not found to be in a factor which was different between groups. This suggests that for these compounds the elevated circulating concentrations are sufficient to allow kidney excretion and therefore urinary concentration is not changed and would not be of great value in defining the disease state. However, many of analytes were changed in the urine which suggests metabolism and excretion changes of these compounds and possible routes for further investigation.

Five factors (Factor 1, 2, 5, 14, 23) were reduced in RD or CaOx stone formation cats. There were a number of analytes that had a relative reduced excretion in the urine that have previously been shown to be elevated in the serum of cats with these conditions. Some examples of this are: 3-(3-amino-3-carboxypropyl) uridine, erythritol, 1-methyladenosine, pseudouridine, 5,6-dihydrouridine, and N1-methylinosine. All of these were decreased in the urine while increased in circulating concentration of cats with CaOx stones [20]. The analyte p-cresol sulfate was elevated in circulation of cats with CaOx stone formation [20] and in RD [27] while decreasing in urine in this experiment. There was an increased circulating concentration and an increased urine concentration of palmitoyl sphingomyelin (d18:1/16:0) [20] in CaOx stone forming cats. Tryptophan circulating concentrations were higher in CaOx stone forming cats and also higher in urine. However, the tryptophan metabolites that are often increased in circulating concentrations in cats with reduced renal function (i.e., kynurenic acid [28]; kynurenate [20]) were not changed in the urine. For these compounds with known circulating concentration elevations but reduced urinary content (both as compared to creatinine) these data suggest that there is reduced renal clearance. This change in fractional excretion may be the result of either another method of excretion (fecal or enhanced degradation to other compounds) or some control of production. This control could be through a feedback loop with the specific compound concentration influencing the enzyme systems producing it. For those compounds where there is an increased concentration in the blood but no difference in the urine, these data suggest that the higher circulating concentrations are effective in allowing normal amounts of excretion in the presence of reduced fractional excretion. It is especially significant that harmful compounds have their circulating concentrations increased and urinary concentration decreased in the diseases where increased circulating concentrations are harmful. Perhaps the most important of this class of compounds that were elevated in the blood but decreased in the urine are the renal toxins. For example, cresol sulfate is a known renal toxin in cats [30,31]. This suggests the need to reduce production of this bacterial derived toxin which can be done through reducing dietary protein [32] or by feeding a food with enhanced betaine and prebiotics [33].

Perhaps most interesting are the analytes that increased in urine as compared to healthy controls. This was seen in Factor 12 which was elevated in the CaOx cats. The purine bases: inosine and hypoxanthine were elevated in the urine. Inosine has been reported to be reduced in RD cats [34] while circulating concentrations were not elevated in CaOx forming cats [20]. Inosine in the urine was similar in healthy cats and cats with renal disease while cats with CaOx stones had greater than a twofold average increase when compared to healthy cats. This suggests that there may be an increased fractional excretion, or excretion is increased based on the direct effect of the stones and may not be the result of renal clearance from blood. It is possible that urinary inosine is increased through a direct effect while urinary xanthine and hypoxanthine are influenced by subsequent degradation and an incomplete conversion to urate, as urate was not elevated in CaOx cats.

Inosine is an important purine biosynthetic intermediate as well as a metabolite in purine degradation. The role of inosine has recently been reviewed [35,36]. Inosine is involved with both activation and suppression of inflammation as a biological signal marker. Interestingly, it was concluded that inosine augmented the production of inflammatory cytokines [37]. These data cannot differentiate between inosine being a promoter of the inflammation associated with CaOx stone formation, or a response generated to suppress this inflammation. Unlike the purine nucleic acid metabolites included in Factor 12 the other purine degradation nucleic acid metabolites (those in factor 1, as well as allantoin and urate) were decreased in cats with RD and intermediate in cats with CaOx stones. The different excretion concentrations of the related purine compounds (inosine and hypoxanthine increased in the CaOx stone formers with only small changes in urate and allantoin) suggest a further area for research. These urinary compounds could also be significant biomarkers of stone formation. However, it is crucial in looking at this association as in all of the changing factors reported here, to note that this does not establish causality and that future prospective controlled studies will be needed to establish if these changes are causal or simply associated in responding similarly to the changing biological milieu of these diseases.

Factor 12 was increased in cats with CaOx stones as compared to healthy cats. The concentration of BHBA was a component of this being numerically higher than healthy controls and cats with RD. There was an observed reduction in circulating concentrations of BHBA between cats with CaOx stones and healthy cats [20]. This suggests that there is a specific loss of BHBA in CaOx stone cats and some of that loss may be accounted for by increased secretion. It is possible that the increased urinary output is the result of increased renal production of BHBA. This would be a biological benefit as BHBA, as an anti-inflammatory signal, could offset the inflammation associated with stone formation. However, it was reported in rats that renal production was first mixed with circulating BHBA before being excreted [38]. In cats with RD and CaOx stones, even with the divergent secretion of BHBA, the reduced circulating concentrations suggests a possible benefit of dietary feeding regimens that would result in an increased circulating concentration of BHBA for cats with CaOx stones or renal insufficiency. It has been reported that a simultaneous dietary increase of medium chain (primarily C8 and C10) fatty acids and very long chain omega-3 fatty acids (DHA and EPA) resulted in a significant increase in BHBA in cats [39]. This possible dietary enhancement of BHBA could prove of value because of a number of positive effects. It has been shown that BHBA protects kidney function (in rats) by decreasing the negative effects of hypertension [40]. This perspective of the value of enhanced circulating ketones is the conclusion of a recent review evaluating the effects of ketone bodies to aid in the management of kidney disease who noted that BHBA changed gene expression and reduced inflammation and oxidative stress [41]. This association supports the continued investigation of dietary factors that increase BHBA in both RD and CaOx stone forming cats.

The etiology of RD and CaOx stone formation is not evaluated by this study. Future research in cats will undoubtedly evaluate genetics, foods consumed and hydration all of which would be expected to influence both initiation and progression of these diseases. Hydration is especially important for cats [29] and a reasonable risk factor for both conditions which should be considered as first priority in this endeavor.

The most significant limitation to this study is that it was limited to colony cats and therefore has a relatively low number of cats in the RD and CaOx group. This limited sample number also limits the epidemiological application of the study. Although the significant results are likely to be applicable to non-colony cats, it is best used as an indicator of possible interventions to aid in the management of these conditions and to design further studies to investigate the control of these compounds. This study is also limited by not having complete 24 hour urine collections so that observed changes in the ratio of analyte to creatinine cannot be directly compared to total excretion. The fact that renal disease starts in small increments and progresses while it is undiagnosed is also a limitation in that this study must then use all of the samples as part of the class of pets with renal disease.

In conclusion the metabolomic changes seen in this retrospective study in the urine of cats with RD and CaOx stones suggests different responses to these conditions. The increased secretion of the purines: inosine, xanthine, and hypoxanthine in cats with CaOx stones suggests a need for an understanding of inosine and its role as a biological compound in

   

inflammation and associated renal changes with CaOx stone formation. The changes in BHBA suggest a possible role for dietary increases in nutrients that increase BHBA to aid in the management of these conditions.

## Supporting information

**S1 Table. Metabolite means and standard errors.**
(XLSX)

**S2 Table. Factor eigenvalues and percent of variance.**
(XLSX)

**S3 Table. Complete factor loadings.**
(XLSX)

**S4 Table. Metabolite data.**
(XLSX)

**S5 Table. Metabolite legends.**
(XLSX)

## Acknowledgments

The authors wish to acknowledge the statistical support provided by Dr. John Bredja.

## Author contributions

**Conceptualization:** Dennis E. Jewell, Kiran S. Panickar.

**Data curation:** Dennis E. Jewell, Selena K. Tavener, Regina L. Hollar, Kiran S. Panickar.

**Formal analysis:** Dennis E. Jewell, Selena K. Tavener, Regina L. Hollar, Kiran S. Panickar.

**Funding acquisition:** Dennis E. Jewell, Kiran S. Panickar.

**Investigation:** Dennis E. Jewell, Selena K. Tavener, Regina L. Hollar, Kiran S. Panickar.

**Methodology:** Dennis E. Jewell, Selena K. Tavener, Regina L. Hollar, Kiran S. Panickar.

**Project administration:** Dennis E. Jewell, Kiran S. Panickar.

**Resources:** Dennis E. Jewell, Kiran S. Panickar.

**Software:** Dennis E. Jewell, Selena K. Tavener, Kiran S. Panickar.

**Supervision:** Dennis E. Jewell, Kiran S. Panickar.

**Validation:** Dennis E. Jewell, Kiran S. Panickar.

**Visualization:** Dennis E. Jewell, Kiran S. Panickar.

**Writing – original draft:** Dennis E. Jewell, Selena K. Tavener, Regina L. Hollar, Kiran S. Panickar.

**Writing – review & editing:** Dennis E. Jewell, Selena K. Tavener, Regina L. Hollar, Kiran S. Panickar.

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
