## [Decision Letter · Decision Letter 0]

13 May 2025

PONE-D-25-17831Urine metabolomic changes in cats with renal disease or calcium oxalate urolithsPLOS ONE

Dear Dr. %LAST_NMAE%,

Thank you for submitting your manuscript to PLOS ONE. After careful consideration, we feel that it has merit but does not fully meet PLOS ONE’s publication criteria as it currently stands. Therefore, we invite you to submit a revised version of the manuscript that addresses the points raised during the review process.

Thank you for submitting the following manuscript to PLOS ONE.

Please revise the manuscript according to the reviewers' comments and upload the revised file.

We look forward to receiving your revised manuscript.

Kind regards,

Yung-Hsiang Chen, Ph.D.

Academic Editor

PLOS ONE

Journal Requirements:

“The authors wish to acknowledge the statistical support provided by Dr. John Bredja and the excellent technical support provided by the animal care technicians of the Hill’s Pet Nutrition feline colony.”

“Funding: Funded by and performed at the Pet

Nutrition Center, Hill’s Pet Nutrition, Inc., Topeka,

KS (http://www.hillspet.com/our-company.html).

The funders had no role in study design, data collection and analysis, decision to publish. The words and conclusions are uniquely the authors and not the funding agency.”

4. Thank you for stating the following in the Competing Interests/Financial Disclosure * (delete as necessary) section:

“We have read the journal's policy and have the following to declare: Three of the authors have an affiliation (K.S.P., S.K.T., and R.L.H) and another formerly had an affiliation (D.E.J.) to the commercial funders of this research, as employees of Hill’s Pet Nutrition, Inc.”

We note that you received funding from a commercial source: “Hill’s Pet Nutrition, Inc.”

6. Please include captions for your Supporting Information files at the end of your manuscript, and update any in-text citations to match accordingly. Please see our Supporting Information guidelines for more information: http://journals.plos.org/plosone/s/supporting-information .

Additional Editor Comments:

Thank you for submitting the following manuscript to PLOS ONE.

Please revise the manuscript according to the reviewers' comments and upload the revised file.

Reviewers' comments:

Reviewer's Responses to Questions

**Comments to the Author**

1. Is the manuscript technically sound, and do the data support the conclusions?

Reviewer #1: Yes

Reviewer #2: Partly

Reviewer #3: Yes

2. Has the statistical analysis been performed appropriately and rigorously? 

Reviewer #1: Yes

Reviewer #2: Yes

Reviewer #3: Yes

3. Have the authors made all data underlying the findings in their manuscript fully available?

Reviewer #1: No

Reviewer #2: Yes

Reviewer #3: No

4. Is the manuscript presented in an intelligible fashion and written in standard English?

Reviewer #1: Yes

Reviewer #2: Yes

Reviewer #3: Yes

5. Review Comments to the Author

Reviewer #1: This is a very interesting and good organized research. Renal disease (RD) and

calcium oxalate (CaOx) stones are most important problems on cat . The objective of this study was to assess urine

metabolomic parameters of cats with these diseases to determine metabolic

abnormalities and differences between the groups. For these reasons;

1. Please give more details about urine and hematological/biochemical results for healthy and sick cats. How did you do the diagnose? Inorganic phosphorus, BUN, Creatinine, urine microscobic analyses, Protein levels, glucose and the position of anemia, WBC and PLT? What about correlations between these parameters and metabolites?

2.Did you do any microbiological analysis on urine and ıf you have them please give the results.

3. Did you know anything about viral infections?

4. Do you have any news about the etiological factors of Renal disease and Calcium oxalate

5. Inclusion and exclusion criterias of your cases? I can not see Table S1 and S2.

6. What about nutrition of your cases?

7. Your introduction is so short. You should more different literatures such as:

The clinical efficacy of cGMP-specific sildenafil on mitochondrial biogenesis induction and renal damage in cats with acute on chronic kidney disease. M Maden, M Ider, ME Or, B Dokuzeylül, E Gülersoy, MC Kılıçkaya, B Bilgiç, ...(2024)

BMC Veterinary Research 20 (1), 499

Clinical efficacy of marbofloxacin in dogs and cats diagnosed with lower urinary tract disorders. B Dokuzeylül, B Celik, BD Siğirci, BB Kahraman, SÜ Saka, A Kayar, S Ak, ..(2019) Med. Weter 75 (9), 549-552

Reviewer #2: PONE-D-25-17831 Review comments

Urine metabolomic changes in cats with renal disease or calcium oxalate uroliths

This well-structured study provides valuable insights into urine metabolomic profiles in cats with renal disease (RD) and calcium oxalate (CaOx) uroliths. The use of repeated urine samples, creatinine normalization, and robust data reduction through factor analysis strengthens the dataset and supports the identification of disease-specific metabolic patterns. The findings are intriguing and suggest that urinary metabolites offer more than a passive reflection of serum changes, potentially revealing functional kidney alterations.

Strengths:

Clear research objectives with appropriate rationale.

Inclusion of both RD and CaOx groups with healthy controls for comparative profiling.

Longitudinal sampling per cat and correct statistical handling of repeated measures.

Application of factor analysis to reduce data complexity and focus on key metabolic features.

Proper normalization of analyte concentrations to urinary creatinine.

Identification of novel metabolic signatures, particularly involving purine nucleotides and phospholipids.

Areas for Improvement:

Disease Characterization: The lack of specific details regarding the type, stage, and severity of RD, as well as the composition of CaOx stones, limits the interpretation and comparability of the findings.

Factor Interpretation and Biological Significance: The biological interpretation of the identified factors and the underlying metabolic pathways needs significant expansion. This includes explaining the elevation of certain metabolites in specific groups and integrating the findings with known feline physiology, diet, and the pathophysiology of RD and urolithiasis in cats and other species.

Graphical Presentation: The inclusion of figures (e.g., heatmaps, volcano plots, pathway enrichment maps) is needed to enhance clarity and engagement.

Causality vs. Association: While the study shows associations, it's crucial to acknowledge that it doesn't establish causality between metabolic alterations and disease pathogenesis.

Terminology and Language: Ensure consistent use of terms (e.g., RD vs. CKD) and address minor grammatical issues.

Reviewer #3: I would like to congratulate the authors on the interesting and relevant study, which raises important hypotheses regarding urinary metabolomic alterations associated with renal disease and calcium oxalate urolithiasis in felines. Below, I present some comments and suggestions.

The attached table was provided in Word format, making it impossible to view all essential clinical data such as age, urinary creatinine, and other parameters.

Line 23: The description of the factors is overly detailed and may hinder understanding for clinical readers. It is suggested to summarize the main findings with a focus on the most relevant clinical/metabolic implications.

Line 45: The small sample size of the reference limits the epidemiological relevance of the study. It is recommended to use additional or more robust sources for context.

Line 50: Although not the focus of the study, struvite and diet are mentioned without addressing dietary mechanisms related to oxalate stones, which are central to the manuscript.

Lines 61–63: It would be enriching to briefly contextualize the role of urinary metabolomics in veterinary medicine, especially in the areas of feline nephrology and urolithiasis. It would be interesting to highlight current knowledge gaps and justify the choice of the metabolomic approach as a diagnostic and research tool.

Line 68: Information on environmental management and inclusion criteria for experimental groups is vague. It is recommended to include details such as type of diet, age range, and diagnostic criteria used for group classification (e.g., imaging, clinical history, laboratory data).

Line 80: There is no detail regarding the method of urine collection, measures to avoid contamination, or average storage time before analysis.

The methodological description relies heavily on external references. It is recommended to specify the number of technical replicates, the use of quality controls, and the criteria for data exclusion.

The “annual and end-of-life” urine collection lacks a more precise definition regarding the timing within the renal disease course. It would be important to detail the intervals between collections and the clinical stage of the animals at the time of sampling.

Line 310: The limitation of the sample to a cat colony restricts the generalization of the findings, considering the low genetic, environmental, and nutritional variability. This aspect could be further emphasized in the study’s limitations.

The manuscript suggests that the increase in urinary inosine in cats with CaOx stones may reflect an inflammatory process. However, inflammatory biomarkers (such as cytokines) that would support this hypothesis were not evaluated. Although the cited literature suggests an immunomodulatory role of inosine, extrapolation to felines remains speculative. A more cautious reformulation of the interpretation is recommended, highlighting the absence of direct evidence.

The proposal that increased urinary BHBA may be beneficial due to its anti-inflammatory effects is interesting, but based on studies in other species (e.g., rats). It is suggested to present this interpretation as a hypothesis to be tested and not as a direct inference, reinforcing the need for controlled clinical studies in cats.

The practice of normalizing urinary concentrations by creatinine assumes stable excretion of this substance. This premise may be questionable in cats with renal dysfunction. It is recommended to present absolute urinary creatinine data between groups to verify the validity of this correction.

6. PLOS authors have the option to publish the peer review history of their article (what does this mean? ). If published, this will include your full peer review and any attached files.

**Do you want your identity to be public for this peer review?** For information about this choice, including consent withdrawal, please see our Privacy Policy .

Reviewer #1: No

Reviewer #2: No

Reviewer #3: **Yes: ** MARIA EDUARDA GONÇALVES TOZATO

---

## [Author Response · Author response to Decision Letter 1]

11 Jul 2025

Journal requirements

1 – we have reviewed the manuscript and believe it meets the PLOS ONE’s style.

2 - This funding statement should meet your needs.

Funding: This research was funded by Hill’s Pet Nutrition, Inc., Topeka, KS, USA (http://www.hillspet.com/our-company.html). Accessed April 27, 2024 which also supplied the excellent technical support of the animal care technicians.

3 – Statements have been updated as requested.

4 – Statements have been updated as requested.

5- all ethics statements are now in then methods section

6 – we have checked the captions and supporting information and it seem correct

We are including the raw data as a supplementary file to make it publicly available at publication

---

## [Decision Letter · Decision Letter 1]

25 Jul 2025

Urine metabolomic changes in cats with renal disease or calcium oxalate uroliths

PONE-D-25-17831R1

Dear Dr. Jewell,

We’re pleased to inform you that your manuscript has been judged scientifically suitable for publication and will be formally accepted for publication once it meets all outstanding technical requirements.

Kind regards,

Yung-Hsiang Chen, Ph.D.

Academic Editor

PLOS ONE

Additional Editor Comments (optional):

Congratulations on the acceptance of your manuscript, and thank you for your interest in submitting your work to PLOS ONE.

Reviewers' comments:

Reviewer's Responses to Questions

**Comments to the Author**

1. If the authors have adequately addressed your comments raised in a previous round of review and you feel that this manuscript is now acceptable for publication, you may indicate that here to bypass the “Comments to the Author” section, enter your conflict of interest statement in the “Confidential to Editor” section, and submit your "Accept" recommendation.

Reviewer #1: All comments have been addressed

Reviewer #2: (No Response)

Reviewer #3: All comments have been addressed

2. Is the manuscript technically sound, and do the data support the conclusions?

Reviewer #1: Yes

Reviewer #2: Yes

Reviewer #3: Yes

3. Has the statistical analysis been performed appropriately and rigorously? 

Reviewer #1: Yes

Reviewer #2: Yes

Reviewer #3: Yes

4. Have the authors made all data underlying the findings in their manuscript fully available?

Reviewer #1: Yes

Reviewer #2: Yes

Reviewer #3: Yes

5. Is the manuscript presented in an intelligible fashion and written in standard English?

Reviewer #1: Yes

Reviewer #2: Yes

Reviewer #3: Yes

6. Review Comments to the Author

Reviewer #1: (No Response)

Reviewer #2: The authors have improved the manuscript in response to earlier comments. The study provides valuable insights into how urine metabolomics can help understand renal disease (RD) and calcium oxalate (CaOx) uroliths in cats.

However, a few clarifications are still needed:

The methods for classifying RD and CaOx are now clearer, but please indicate whether the CaOx stones were pure calcium oxalate or mixed types, if known.

Please explicitly state in the conclusion that the findings represent associations rather than causative relationships, and that further studies are needed to confirm the biological roles of these metabolites.

Recommendation: Accept with Minor Changes

The study is scientifically sound and well-executed. With these small clarifications—particularly in the conclusion and disease characterization—it will be suitable for publication.

Reviewer #3: All the concerns raised in the previous round have been adequately addressed. I have no further comments.

7. PLOS authors have the option to publish the peer review history of their article (what does this mean? ). If published, this will include your full peer review and any attached files.

**Do you want your identity to be public for this peer review?** For information about this choice, including consent withdrawal, please see our Privacy Policy .

Reviewer #1: No

Reviewer #2: No

Reviewer #3: **Yes: ** MARIA EDUARDA GONÇALVES TOZATO

---

## [Editor Report · Acceptance letter]

PONE-D-25-17831R1

PLOS ONE

Dear Dr. Jewell,

I'm pleased to inform you that your manuscript has been deemed suitable for publication in PLOS ONE. Congratulations! Your manuscript is now being handed over to our production team.

Kind regards,

on behalf of

Dr. Yung-Hsiang Chen

Academic Editor

PLOS ONE